# Effects of spectral manipulations of music mixes on musical scene analysis abilities of hearing-impaired listeners

**Aravindan Joseph Benjamin**[1]*, Kai Siedenburg[1,2]

**1** Dept. of Medical Physics and Acoustics, Carl von Ossietzky University of Oldenburg, Oldenburg, Germany,
**2** Signal Processing and Speech Communication Laboratory, Graz University of Technology, Graz, Austria

* aravindan.benjamin@uni-oldenburg.de

**Data Availability Statement:** The data supporting the findings of this work are available in the following link at: https://doi.org/10.5281/zenodo.14541533.

## Abstract

Music pre-processing methods are currently becoming a recognized area of research with the goal of making music more accessible to listeners with a hearing impairment. Our previous study showed that hearing-impaired listeners preferred spectrally manipulated multi-track mixes. Nevertheless, the acoustical basis of mixing for hearing-impaired listeners remains poorly understood. Here, we assess listeners' ability to detect a musical target within mixes with varying degrees of spectral manipulations using the so-called EQ-transform. This transform exaggerates or downplays the spectral distinctiveness of a track with respect to an ensemble average spectrum taken over a number of instruments. In an experiment, 30 young normal-hearing (yNH) and 24 older hearing-impaired (oHI) participants with predominantly moderate to severe hearing loss were tested. The target that was to be detected in the mixes was from the instrument categories Lead vocals, Bass guitar, Drums, Guitar, and Piano. Our results show that both hearing loss and target category affected performance, but there were no main effects of EQ-transform. yNH performed consistently better than oHI in all target categories, irrespective of the spectral manipulations. Both groups demonstrated the best performance in detecting Lead vocals, with yNH performing flawlessly at 100% median accuracy and oHI at 92.5% ($IQR = 86.3$–96.3%). Contrarily, performance in detecting Bass was arguably the worst among yNH ($Mdn = 67.5\%$ $IQR = 60$–75%) and oHI ($Mdn = 60\%$, $IQR = 50$–66.3%), with the latter even performing close to chance-levels of 50% accuracy. Predictions from a generalized linear mixed-effects model indicated that for every decibel increase in hearing loss level, the odds of correctly detecting the target decreased by 3%. Therefore, baseline performance progressively declined to chance-level at moderately severe degrees of hearing loss thresholds, independent of target category. The frequency domain sparsity of mixes and larger differences in target and mix roll-off points were positively correlated with performance especially for oHI participants ($r = .3$, $p < .01$). Performance of yNH on the other hand remained robust to changes in mix sparsity. Our findings underscore the multifaceted nature of selective listening in musical scenes and the instrument-specific consequences of spectral adjustments of the audio.

**Funding:** Freigeist Fellowship from the Volkswagen Stiftung.

**Competing interests:** The authors have declared that no competing interests exist.

## Introduction

What makes a good musical mix for listeners with a hearing loss? This seemingly straight-forward question hosts a plethora of questions in music production and psychoacoustics that research is only beginning to address. In fact, sounds from musical instruments tend to overlap substantially in time and frequency in music mixes and listeners with different hearing abilities (and of potentially different age ranges) vary in their ability to separate sound sources percep-tually [1]. Yet, little is known about how properties of the mix affect this process. Neither do we have substantiated knowledge on whether different groups of listeners such as listeners with and without hearing loss react in similar or dissimilar ways to manipulations of a musical mix. Here, we attempted to approach these questions by devising an experiment based on selective listening in musical mixtures, wherein listeners were tasked to detect a cued target sound in musical mixtures that were manipulated in the frequency domain.

Selective listening has been a thoroughly researched field within the context of auditory scene analysis (ASA) [2]. A very large part of research dealing with selective attention has been performed on the so called *'cocktail party problem'* [3]. Here, the perceptual processes in a receiver involved in tracking and understanding one speaker amid competing speakers in a setting similar to a cocktail party are of focus [4]. The terminology was coined by Collin Cherry who initially conducted experiments where participants were tasked separating differ-ent speech signals presented diotically [5]; he showed that separability of the signals in the presence of background noise depended upon the rate of the speech, its direction of arrival, the participants' gender, and average pitch of the speech signals. However, given the focus on speech perception, the task of detecting and tracking musical targets in the presence of accom-panying musical maskers remains underexplored (i.e., Musical Scene Analysis or MSA tasks), especially within the context of sensorineural hearing impairment.

### Previous work

The study by Siedenburg et al. [6] was among the first to investigate MSA ability as a function of hearing impairment. Here, melody and timbre discrimination abilities of a sample of young normal-hearing and older hearing-impaired listeners were investigated. In the melody dis-crimination task, first a reference signal which was a clarinet target along with Piano, Cello, or noise maskers was presented. This was followed by two, one second Clarinet excerpts with varying pitch sequences. The participants were tasked with detecting which one of the two Clarinet excerpts was in the reference. Similarly, in the timbre task, target and masker refer-ences with the same maskers were presented and followed by successive Trumpet or Flute excerpts with identical pitch sequences. Here, the participants had to determine which of the two instruments was in the reference. It was shown that the older hearing-impaired partici-pants required signal-to-masker ratios 10 dB greater on average compared to the young nor-mal-hearing participants. However, musical training among both groups brought about a reduction to these requirements. The older hearing-impaired listeners also did not utilize level drops in the maskers in both tasks unlike normal-hearing listeners. In a later study by Sieden-burg et al. [7], four target musical voices taken from Johann Sebastian Bach's *'The Art of the Fugue'* were presented through four spatially separated loudspeakers. The number of voices presented at a given time was varied between two and four. The cued voice was played first, and a tremolo by way of amplitude modulation was applied or not applied in a given trial and presented through one of the four loudspeakers along with the other voices. The participants were then tasked with detecting if a tremolo was present in the cued voice in the subsequent presentation. It was observed that the timbral homogeneity of the instrumentation, directly

related to spectral similarity of the target sounds with the accompanying voices. This in turn had detrimental effects on performance compared to a heterogeneous instrumentation.

In a study investigating MSA abilities among normal-hearing participants, Bürgel et al. [8] showed that Lead vocals prevailed in attracting auditory attention when presented amid coherent multi-track mixes of popular music. Furthermore, detection performance of the participants depended on the order in which the target and the mix were presented in the trials and the target instrument category (e.g., Lead vocals, Guitar, Piano etc.). In a follow up study [9], when frequency micro-modulations of Lead vocals were applied to other target instruments in a similar task, the effect of presentation order seen previously was mitigated, suggesting a role for frequency micro-modulations in auditory salience. More recently, Hake et al. [1] developed a viable open source tool to assess MSA ability with the aid of a scoring system for listeners with varying degrees of hearing impairment. Here it was shown using a large sample of normal-hearing and hearing-impaired participants that detecting the presence of a cued target instrument in a mixture depended greatly upon the level ratios between the target and the mixture, the complexity of the mixture (i.e., the number of instruments), and the target instrument category. Stereo panning width of the individual instruments in the mix however had a smaller impact on the task performance. More importantly, they showed that hearing-impaired listeners with severe to profound hearing loss had significantly lower MSA abilities. However, the effect of spectral changes of the instruments in the mixture were not assessed.

## Motivation

Despite the alarming rise in individuals living with sensorineural hearing impairment as per the WHO projection of over 900 million people by year 2050 [10], research into music processing methods among them remains surprisingly scarce. The music industry relies heavily upon mixing and mastering practices of trained professionals. Here, the so called mixing engineer is required to combine raw recordings of a myriad of instruments and vocals made available through separate tracks into a coherent mixdown or mix, which has been subjected to meticulous spectral and temporal manipulations [11]. The mix unlike its raw constituents should bear optimal transparency of the individual tracks while matching the aesthetic intentions of the artists.

With regards to mixing preferences, Nagathil et al. [12] showed that cochlear-implant (CI) users preferred reduced spectral complexity in classical chamber music which was accomplished through a low-rank approximation of its constant-Q-transform. Using a music re-mixing task, Hwa et al. [13] showed that CI users preferred an average increase of 7.1 dB in bass frequencies and 6.7 dB in treble frequencies compared to the original stimuli. The bass and treble frequencies were defined as those below and above the 50th percentile of the frequencies in the stimuli, respectively. Nevertheless, in a more recent study by Althoff et al. [14], which utilized a similar remixing task, no significant differences in low/high pass filtering preferences for instrumental music were observed between CI users and normal-hearing controls. However, Benjamin and Siedenburg [15] showed that spectral manipulations via the transformed equalization or EQ-Transform performed on constituent tracks of multi-track mixes, increased their objective frequency-domain sparsity by way of higher Gini indices. Importantly, hearing aid users, with mild to moderate hearing loss preferred mixes with spectrally sparser tracks. Sparser time-frequency representations have been postulated at overcoming the cocktail party problem [16]. Furthermore, the effectiveness of source separation algorithms, improves for sparse representations of music [17]. From a psychoacoustical standpoint, cochlear hearing impairment has been shown to give rise to broader auditory filters and reduced frequency selectivity which may in turn depreciate the ability to separate sounds

closer in frequency [18, 19] even when the impairment was mild [20]. This assertion was underpinned in speech perception by Gaudrain et al. [21] where normal-hearing listeners under simulated hearing loss performed better at identifying the order of a vowel sequence when the component vowels were smeared in the frequency domain. Importantly, studies by Lentz and Leek [22] and Narne et al. [23] showed that hearing-impaired listeners have a reduced ability to perceive changes in spectral shape.

Therefore, the specific question addressed by this study is whether spectral manipulations to music can be employed as effective means to enhance scene analysis performance among hearing-impaired listeners. To investigate this question within the context of music perception, we aim at assessing the ability of a participant at detecting a musical target in the presence of musical maskers in a mix, as a function of their level of hearing loss, musical training, and the degree of spectral manipulation applied to both target and the accompanying maskers in the mix. In other words, this work aims at assessing how alterations to the power spectral variation of popular music, affect the MSA abilities of listeners with cochlear hearing loss in multi-track musical scenes. Based on the implications of the EQ-transform on hearing-impaired listeners as shown in [15], it will be used to bring about such alterations in this work. To control for other effects on MSA as demonstrated by Hake et al. [1], complexity and level differences between target and mix were kept constant. For simplicity, we will refer to [15] as our earlier work throughout this manuscript.

## Methods

In this section we outline the participants recruited for the study and the equipment and stimuli used. We then describe the experiment design used to assess MSA performance among the participants. Furthermore, a brief overview of the EQ-transform will be given.

### Participants

A sample of 30 young normal-hearing (yNH) participants and 24 older hearing-impaired (oHI) participants took part in this study. The yNH were recruited using an advertisement posted on-line and oHI were mostly recruited via Hörzentrum Oldenburg gGmbH (Oldenburg, Germany). The recruitment process started on the 9th of May, 2023 and ended on the 20th of September, 2023. All of the participants provided their informed consent in writing by signing a consent form, provided immediately upon participation. The participants could choose either a German or an English version of the consent form. After receiving their informed consent, we proceeded to assess their level of musical training using the Gold MSI musical training questionnaire (musical training subscale) proposed by Müllensiefen et al. [24]. Based on the assessment, the normal-hearing participants were significantly more musically trained than the hearing-impaired participants, $t(52) = 2.1$, $p = .04$, $d = 0.6$ (Medium effect). However, as apparent later in our analysis, musical training had no effect on MSA performance. Afterwards, the hearing loss levels (HL) using pure-tone audiometry at 125 Hz, 250 Hz, 500 Hz, 1 kHz, 2 kHz, 4 kHz, and 8 kHz frequencies were assessed for each of the participant using a portable audiometer for both ears. The mean hearing loss level (MHL) which was an arithmetic mean taken over all of the frequencies over both ears was used to categorise the participant groups as per the guidelines outlined in [25]. Normal hearing were so classified with a MHL $\leq$ 25 dB ($M$ = 6 dB HL, $SD$ = 5.3 dB HL). Among the 24 oHI participants, three of them were classified as having mild hearing impairment with 25 dB < MHL $\leq$ 40 dB ($M$ = 35.2 dB HL, $SD$ = 3.04 dB HL) and 21 of them had moderate to severe hearing impairment or MHL > 40 dB ($M$ = 46.3 dB HL, $SD$ = 4.3 dB HL). However, in this study, all of the oHI participants were grouped together ($M$ = 45 dB HL, $SD$ = 5.6 dB HL). Fig 1(A) shows

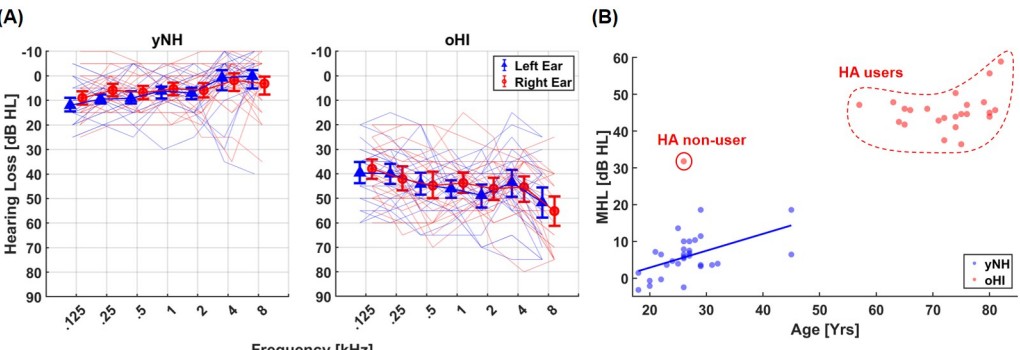

**Fig 1.** (A) Audiograms of the participant groups considered in this study mean hearing loss plotted in thick lines about bootstrapped 95% confidence intervals for the pure tone frequencies considered. The thin lines indicate individual audiograms. (B) The relationship between mean hearing loss level (MHL) and participants' ages. The significant linear correlation among the yNH is shown with a straight line. Among oHI participants, bilateral hearing aid (HA) users and the non-user are highlighted.

an illustration of the hearing loss levels at the aforementioned frequencies for both participant groups. It can be observed from Fig 1(A) that the mean hearing loss among the hearing-impaired participants is around 40 dB greater than that of the normal-hearing participants.

The normal-hearing participants who took part in this study were relatively young with ages of 18 and 45 years ($M$ = 26.6 yrs, $SD$ = 6.1 yrs). The hearing-impaired participants on the other hand were significantly older and were between the ages of 26 and 82 years ($M$ = 71 yrs, $SD$ = 11.6 yrs). As all except one among the hearing-impaired participants were above the age of 50 years, in this study we use the abbreviation oHI (older hearing-impaired) to refer to this group. Among the oHI, 22 participants were bilateral behind-the-ear type hearing aid users with only one participant using bilateral in-the-ear type hearing aid. The last remaining participant had no history of hearing aid use. Fig 1(B) illustrates the linear relationship between the participant ages and their respective MHL. As illustrated, a significant linear correlation between age and hearing loss is apparent among yNH with a positive correlation between age and mean hearing loss $r(28)$ = .5, $p$ = .003 < .01. Hearing impaired participants showed a similar correlation only when considered altogether $r(22)$ = .5, $p$ = .006 < .01. However, when we disregard the youngest participant (i.e., < 50 yrs) with no history of hearing aid use, the correlation becomes non-significant ($r(21)$ = .26, $p$ = .2).

## Stimuli and apparatus

All the audio excerpts used as stimuli were 2 seconds long and were taken from the Medley database [26]. The broadband target to mix level ratios were always kept constant at -10 dB. The number of vocals / instrument tracks in all of the mixes were kept at 5. The stimuli playback was presented over a pair of activ 8" near-field studio monitors by ESI Audiotechnik GmbH (Leonberg, Germany) in a low reflection chamber at the University of Oldenburg, Germany. The monitors were separated by a 90° angle and 2 m distance from the listener's seat. The overall playback level was calibrated to be 80 dBA (i.e., A-Weighted equivalent continuous sound pressure level over a measurement duration of 1 minute) at the participant position. This calibration was performed using white noise colored by the ensemble average spectrum of commonly occurring instrument classes and Lead vocals available in the Medley database. The processing on the stimuli were realized using a standalone desktop computer running MATLAB R2023a. This standalone machine was connected to the monitors with the aid of an

RME Fireface UFX audio-interface. The puretone audiometry performed on all of the participants in this study was fulfilled with a portable AD528 audiometer from Interacoustics (www.interacoustics.com/ad528).

## Procedure

The experiment consisted of a detection task similar to that used by Hake et al. [1]. Here, the two-second target sound was presented first and after a second of silence, the two-second mix was presented. All stimuli were presented via a pair of loudspeakers in such a manner that the signals generated from both speakers were identical to avoid the influence of spatial cues on the detection task [27]. All oHI participants using hearing aids were explicitly requested to take them off before the experiment.

The participant upon listening to the two excerpts, was tasked with identifying if the target sound was present in the mix through being prompted to click 'Yes' or 'No' to whether they heard the target in the mix presented to them as illustrated in Fig 2(A). In any given trial where a target sound followed by a mix is presented, the target sound was taken from one of 5 different instrument or target classes: Lead vocals (Lead), Bass guitar (Bass), Drums, Guitar, and Piano. Altogether, 200 trials with distinct target and mix combinations were presented for the five target classes with 40 per target category. Within the 40 trials per target category, 20 trials contained the target and the remaining 20 trials did not contain the target. The 20 trials were then ramified into 4 sets of 5 trials with each set being subjected to a specific % EQ-transform (% EQ Tran) among the ones considered (i.e., 0%, 100% / Factory, 200%, and 300%). Fig 2(B), provides an illustration of EQ-transform and their implications on the power spectrum of a track to which the transform is applied. The transform estimates the transformed power spectrum by way of a linear extrapolation between the original or factory power spectrum (100%) and a reference (0%). The reference is an ensemble average spectrum taken over a number of tracks from a variety of instrument classes. The reference power spectrum is

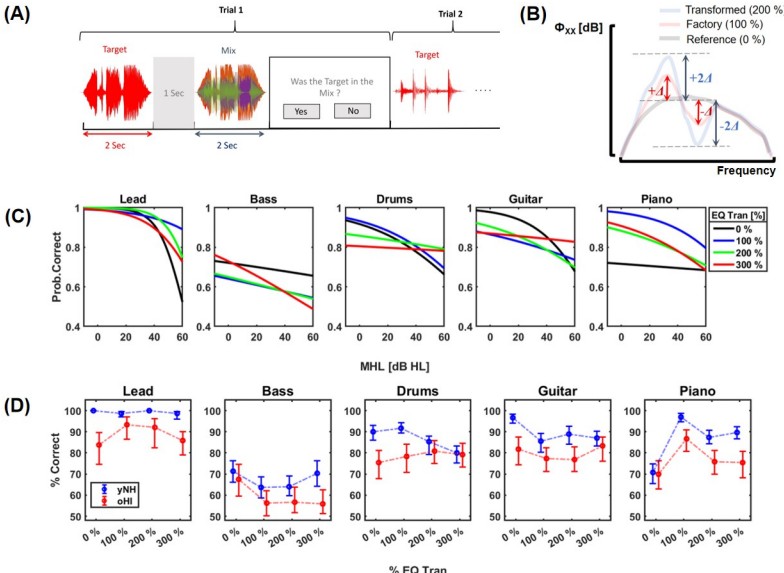

**Fig 2.** (A) Procedure of the experiment. (B) The effect of 200% EQ-transform on the power spectrum. (C) The conditional probability plots illustrating the model output of probability correct for different target classes with respect to mean hearing loss and (D) means about bootstrapped 95% confidence intervals of correct answers as a percentage of the total trials per target category and % EQ-Transforms considered.

always energy normalized with respect to the factory spectrum undergoing the transform. By applying the 200% EQ-transform on a track as shown in Fig 2(B), we essentially double the power level differences between the factory power spectrum and the reference in the transformed spectrum. A 300% transform would commensurately triple this difference and so on. In our earlier work, we showed that for tracks taken from different instrument categories, frequency-domain sparsity sees a monotonic rise with increasing % EQ-Transform.

The % EQ-transform parameters used in the experiment were specifically chosen for they cover the range of those preferred by both yNH and oHI participants studied in our earlier work. The same % EQ-transform was applied to both the target and each track of the corresponding mix in a given trial. The order of the trials were randomized irrespective of target category, presence of the target in the mix, or % EQ-transform applied to avoid order biases in the trials presented between the participants.

## Data analysis

A non-parametric approach was adapted here by way of evaluating bootstrapping sample means using $10^3$ realizations with replacement [28]. A generalized linear mixed-effects model (GLME) was fitted on the data to estimate the main effects of the EQ-transform, mean hearing loss (MHL), the target instrument category, and the interaction between these effects on the dichotomous correct or incorrect responses to the trials. The model also accounted for the interaction between musical training scores and MHL. Furthermore, the model considered random intercepts to account for variability among the participants and the individual trials. The GLME was used as the analysis was performed on the dichotomous correct or wrong answer to those acquire from the participants [29]. Post-hoc analyses were conducted using the Mann-Whitney-U test [30].

## Results

Considering the proportion of correct responses over the total number of trials as a percentage (% Correct), overall median performance of yNH at 84.9% accuracy was around 5% better than oHI at 79.6%. Both groups performed best when detecting lead vocals with yNH performing at a perfect 100% median accuracy and oHI at 92.5%. The worst performance was observed for Bass targets where, both yNH (67.5%) and oHI (60%) came closest to performing at a chance-level of 50%. Nevertheless, yNH consistently outperformed oHI across all target classes. The latter notwithstanding, both yNH and oHI were most accurate at factory settings (100% EQ Tran), with median % correct responses of 88% and 81%, respectively. Fig 2(D) shows the distribution of % correct responses of yNH and oHI, under the different test conditions.

Here, we summarize the output of the GLME model fitted on our data, as classical ANOVA statistics for simplicity. These statistics are derived using MATLAB's `anova` function. Based on the model, musical training had neither an independent effect on performance ($p$=.98), nor did it interact significantly with MHL ($p$=.33). However, there was a main effect of MHL on performance $F(1,10434) = 24.4$, $p < .0001$, for which the model estimates an odds ratio of OR = 0.97 (95% CI: 0.96-0.98). This means, for every unit increase in MHL (+1 dB HL), the model predicts a modest albeit progressive 3% drop in the odds of correctly detecting the target in the mix. The progressive effect of hearing loss on MSA performance can be observed in the conditional probability plots derived from the model, shown in Fig 2(C). Furthermore, while a significant main effect of target category was observed, $F(4,10434) = 30.6$, $p < .0001$, % EQ Tran did not independently affect performance ($p$=.07). Nevertheless, % EQ Tran and MHL interacted significantly, $F(3,10434) = 3.2$, $p = .02 < .05$. A significant two-way interaction

effect between MHL and target category was also observed $F(4,10434) = 11.7$, $p < .0001$ as well as % EQ Tran and target category, $F(12,10434) = 3.8$, $p < .0001$. Lastly a significant three-way interaction on performance was observed between target category, % EQ Tran, and MHL $F(12,10434) = 5.2$, $p < .0001$.

Post-hoc analysis was conducted on the % correct responses accrued over the respective test conditions for both participant groups. As suggested by the model, barring the effects of EQ-Transform and target category, % correct responses among yNH ($Mdn = 84.9\%$, $IQR = 82.9–89.3\%$) were significantly higher than oHI ($Mdn = 79.6\%$, $IQR = 70.3–84.3\%$), $U = 589$, $p < .0001$, $r = .54$ (Very large effect). The model nevertheless suggests a progressive decline in MSA performance with increasing hearing loss. From a baseline performance which was that taken for yNH at 84.9% for a MHL of 6 dB HL and an estimated OR of 0.97 for the main effect of MHL, the model projects that the performance may fall to even chance-levels at an MHL of around 63 dB HL, characterized by moderately severe hearing loss. The negative effect of hearing loss on MSA performance was noticeable across all target classes. Interestingly, both yNH and oHI showed the highest accuracy in detecting Lead vocals. Most notably however, Lead vocals brought on the largest disparity in performance between the groups, with yNH having a perfect median score of a 100%, markedly outperforming the oHI ($Mdn = 92.5\%$, $IQR = 86.3–96.3\%$), $U = 651$, $p < .0001$, $r = 0.73$ (Huge effect). In contrast, both groups performed worst at detecting Bass targets. For the latter, although yNH ($Mdn = 67.5\%$, $IQR = 60–75\%$) performed significantly better than the oHI ($Mdn = 60\%$, $IQR = 50–66.3\%$), the performance gap was the smallest observed across the target classes, $U = 507$, $p = .01 < .05$, $r = 0.3$ (Medium effect). After Lead vocals, Guitar targets elicited the best performance among yNH ($Mdn = 91.5\%$, $IQR = 85.8–94.4\%$). The difference in performance compared to oHI ($Mdn = 81.6\%$, $IQR = 74.5–87.2\%$) was also observably the second largest, $U = 573$, $p = .0002 < .001$, $r = 0.5$ (Large effect). Detection of Piano targets also saw a superior performance in yNH ($Mdn = 86.8\%$, $IQR = 81.7–92\%$) which was significantly better than oHI ($Mdn = 78.1\%$, $IQR = 71.7–85.7\%$), $U = 550$, $p = .001 < .01$, $r = 0.45$ (Large effect). Both groups performed similarly for Drums: yNH ($Mdn = 87.5\%$, $IQR = 82.5–90\%$), oHI ($Mdn = 78.8\%$, $IQR = 75–85\%$), $U = 531$, $p = .003 < .01$, $r = 0.4$ (Large effect).

As suggested by the model, there were no significant main effects of EQ-Transform on the performance among both yNH and oHI ($p > .2$). Nevertheless, the performance disparity between the two groups was most pronounced at factory settings (100% EQ Tran), where yNH ($Mdn = 87.7\%$, $IQR = 85.3–90\%$) significantly outperformed oHI ($Mdn = 80.7\%$, $IQR = 74.3–84.8\%$), $U = 600$, $p < .0001$, $r = 0.6$, (Very large effect). This difference was smallest for 0% settings, with yNH ($Mdn = 85.1\%$, $IQR = 83.1–89.3\%$) still performing significantly better than oHI ($Mdn = 76.8\%$, $IQR = 70.7–86.3\%$), $U = 538$, $p = .001 < .01$, $r = 0.4$ (Large effect). This observation was also made at 200% settings with yNH ($Mdn = 85.8\%$, $IQR = 81.6–88\%$) performing significantly better than oHI ($Mdn = 78.6\%$, $IQR = 71.4–81.9\%$), $U = 576$, $p < .0001$, $r = 0.51$ (Large effect). Performance at 300% settings among both groups was comparable to that shown for 200% with yNH ($Mdn = 85.8\%$, $IQR = 81.6–89.8\%$) performing similarly better than oHI ($Mdn = 78.9\%$, $IQR = 69.6–83.7\%$), $U = 567$, $p = .0002 < .001$, $r = 0.49$ (Large effect).

Factoring in the interaction effect of target category and % EQ Tran, a flawless median performance of 100% was observed for yNH across all % EQ Tran settings for Lead vocals. As such, yNH performed significantly better across all settings except at factory settings where oHI ($Mdn = 100\%$, $IQR = 90–100\%$), performed similarly well. At 300% EQ Tran, performance of oHI ($Mdn = 90\%$, $IQR = 80–95\%$) saw the largest deviation from that of yNH, $U = 602$, $p < .0001$, $r = 0.7$ (Very large effect). Although the disparity in performance for Bass was relatively smaller across all settings, 300% EQ Tran similarly brought about the largest deviation in the performance between yNH ($Mdn = 70\%$, $IQR = 60–80\%$) and oHI ($Mdn = 50\%$, $IQR = 50–$

65%), $U = 535$, $p = .001 < .01$, $r = 0.42$ (Large effect) with the latter performing at almost chance-level. Performance of oHI was similarly close to chance-level at both factory settings ($Mdn = 60\%$, $IQR = 45–65\%$) and 200% ($Mdn = 55\%$, $IQR = 45–65\%$). For Drums, performance of oHI ($Mdn = 80\%$, $IQR = 70–90\%$) remained consistent over all settings while yNH performed best at factory settings ($Mdn = 90\%$, $IQR = 90–100\%$) and 0% ($Mdn = 90\%$, $IQR = 80–100\%$). The largest disparity in performance for Drums was observed between the groups for the latter setting, $U = 560$, $p = .0002 < .01$, $r = 0.5$ (Large effect). For Guitar, yNH ($Mdn = 89\%$, $IQR = 77.8–100\%$) performed similarly across all settings except at 0% settings ($Mdn = 100\%$, $IQR = 87.5–100\%$), where they performed best. As such, the performance disparity compared to oHI ($Mdn = 87.5\%$, $IQR = 75–93.8\%$), was observably the largest $U = 574$, $p < .0001$, $r = 0.6$ (Very large effect). Across all settings except 300%, where performance of oHI ($Mdn = 89\%$, $IQR = 78–89\%$) was comparable, yNH performed significantly better. On the other hand, for all settings for Piano except 0%, yNH performed significantly better. This effect was most pronounced at factory settings ($Mdn = 100\%$, $IQR = 90–100\%$) where the largest deviation in performance from oHI ($Mdn = 90\%$, $IQR = 85–90\%$) was observed, $U = 566$, $p < .0001$, $r = 0.5$ (Very large effect).

In order to assess if a statistical trend in performance of oHI existed by virtue of a step-wise increase in % EQ-Tran, we conducted the Jonckheere-Terpstra test [31]. The test was conducted on the % correct responses across the four degrees of spectral manipulation, going from 0% EQ Tran implying the lowest spectral contrast, to 300% implying the highest. By doing so, a significant, monotonically decreasing trend in the performance of oHI in detecting Bass targets was observed, $J\text{-}T = 1416$, $Z = -2.08$, $p = .03 < .05$, $\tau = -.172$ (Medium effect). For the other target classes however, no such trend could be shown for oHI.

Although it can be observed from Fig 2(D), that manipulating the spectral constrast using the EQ-Transform does bring about changes to the performance, neither of the participant groups saw any improvement in their MSA performance with over-mixing (EQ Tran > 100%). In spite of these observations, it should be acknowledged that alterations to contrast may affect objective spectral descriptors, such as the frequency-domain sparsity of the stimuli nevertheless. However, as previously mentioned, the different % EQ Tran were not applied on the same tracks for an objective comparison in this work. Therefore, it cannot be ascertained that increasing the degree of spectral manipulations will give rise to higher objective frequency domain sparsity through higher Gini indices, because the global energy densities of the tracks may well vary. As shown in Fig 3, although there are marginal changes in Gini indices for different % EQ Tran, neither the target nor the mix sparsity saw a significant increase ($p > .18$) by virtue of the higher degrees of spectral manipulations.

Apart from the Gini index, the spectral roll-off point (roll-off) was considered as another objective descriptor. This spectral descriptor provides the upper frequency limit below which 95% of the energy of the signal is contained, thereby indicating the rate at which the energies decay over frequency. Among other applications, this descriptor has been used widely within the context of genre classification [32] in music and at discerning speech from music [33]. The overall effect of EQ-transform for this descriptor as shown in Fig 3 was significant only for the mix, $\chi^2(3) = 68$, $p < .0001$, $\eta^2 = 0.5$ (Large effect). Moreover, a significant monotonical increase in the roll-off of the mixes with increasing % EQ-transform $p < .05$ was evident overall. The $p$-values reported herein were subjected to Bonferroni-Holm corrections to account for family-wise errors in multiple comparisons [34].

Unlike in the case of the Gini indices, the spectral manipulations through the EQ-transform do bring about changes to the roll-off points of the mixes. However, the question beckons if these spectral descriptors on their own, influence MSA performance among the participant groups. We therefore assessed a linear regression between the two descriptors and the MSA

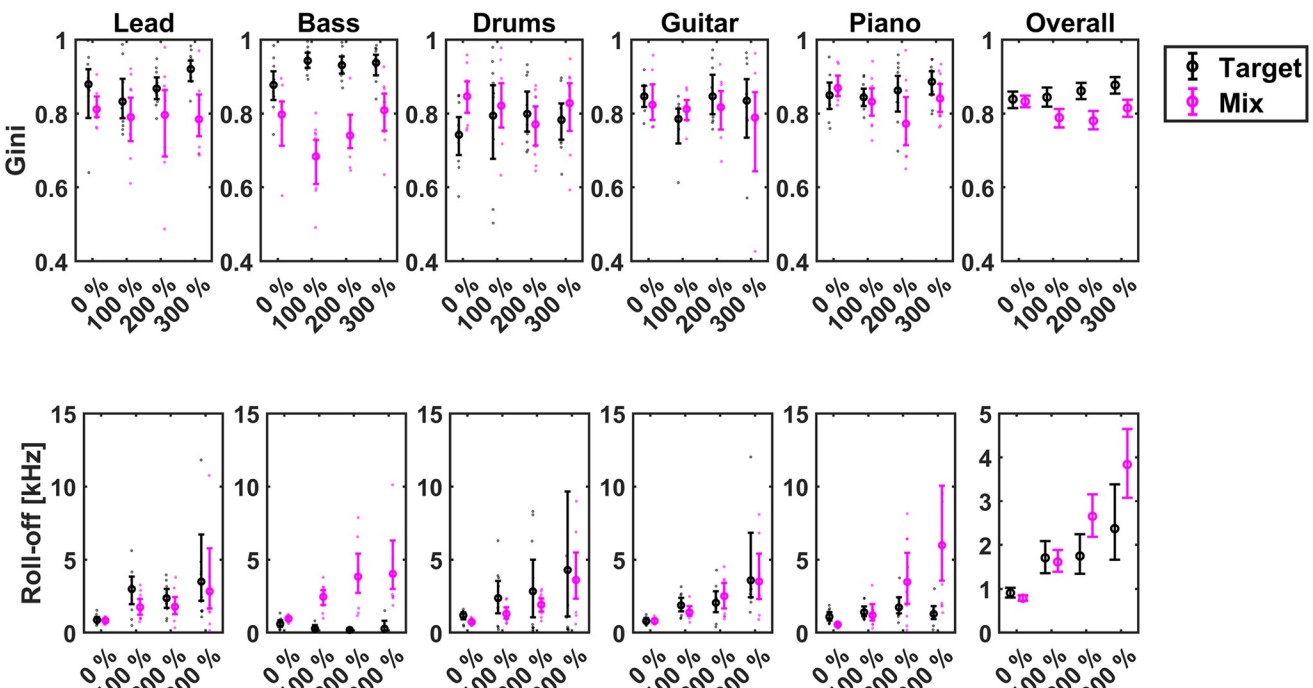

**Fig 3. 95% confidence interval plots about boot-strapped means illustrating the effects of % EQ-transform on the Gini indices and spectral roll-off points of the targets and mixes considered.**

performance. Here, the % correct was that evaluated over all the participants in a given group (i.e. yNH and oHI) for the answers accrued over the individual trials bearing distinct values for the descriptors. Fig 4 gives an illustration of the linear correlation between the aforementioned descriptors and the % correct answers for the two groups. It is apparent that as the mixes became objectively sparser in the frequency domain, oHI participants saw an improvement in their performance $r = .3$, $p = .001 < .01$ as shown in Fig 4(A). As the roll-off does not indicate any transparency markers of the target amidst the mix, we took the vector difference between roll-off of the target and the masker (i.e. target roll-off—mix roll-off). In such a difference, a positive quantity indicates that the energy of the mix is distributed mostly over a smaller bandwidth than that of the target. A positive linear correlation between this roll-off difference and % correct answers is shown in Fig 4(B) among both the yNH $r = .3$, $p = .0001 < .001$. and the oHI $r = .3$, $p = .0003 < .001$. This may indicate that the narrower the range of frequencies over which the energy distribution of the mix becomes relative to that of the target, the less the target is energetically masked. Lastly, it was shown that musical training showed no significant correlation ($p > .3$) with MSA performance among neither participant groups. This assertion is supplemented by the model predictions outlined earlier where musical training had no effect on MSA performance.

## Discussion

The findings of our detection task improve our understanding of a number of factors influencing the ability of listeners of varying degrees of hearing impairment to detect a target track within a musical mix. Notably, the type of instrument in the target track or target category emerged as a strong determinant of performance. Both participant groups performed best at

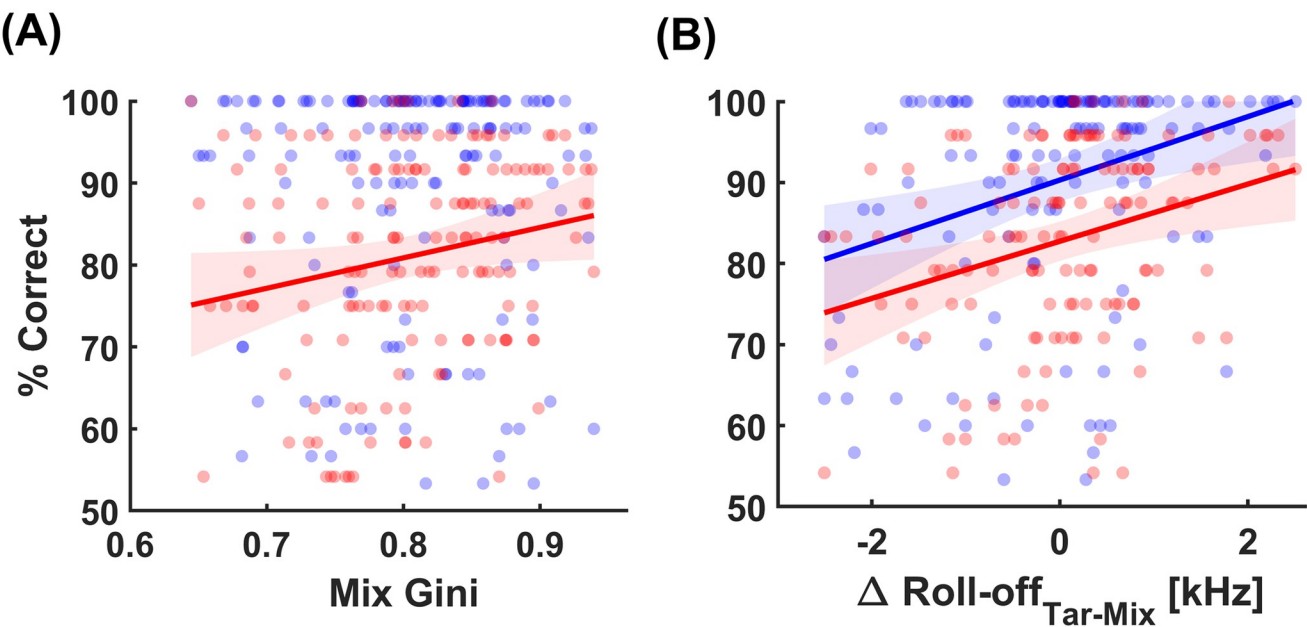

**Fig 4. Linear correlations Gini index of the mixes (A), roll-off differences between target and mix (B) and % correct.** Blue markers and trend lines indicate correlation among yNH and red oHI participants. Shaded regions shown with the trend lines correspond to the 95% confidence intervals.

detecting Lead vocals while demonstrating the worst performance for Bass, so much so that oHI participants performed at near chance-levels. This suggests that the perceptual salience of different instruments plays a pivotal role in the overall task performance. The level of hearing loss among participants demonstrated a modest negative influence on their detection ability. As such, MSA performance progressively declined with increasing hearing loss, reaching even chance-levels of performance at thresholds associated with moderately severe hearing impairment. The degree of spectral manipulation, while enhancing mixing preferences among oHI [15], was found to have no noticeable effect on their MSA performance. Surprisingly, higher degrees of spectral manipulation even depreciated the performance of oHI in detecting Bass targets. In [35], we showed that although spectral manipulations to musical scenes did not confer benefits to MSA performance, the subjective sound quality ratings were heavily influenced by the degree of the manipulations, especially in yNH. Interestingly, MSA performance strongly correlated with the quality ratings in oHI participants. This suggests that improved scene analysis abilities among listeners with hearing loss, may have a favorable influence on their overall listening experience of multi-track music.

Furthermore, our results indicate that spectral manipulations on both the target and mix had varying degrees of influence on MSA performance. Unlike that of the target, the Gini indices of the mixes were associated with MSA performance of oHI participants, enhancing their ability to discern the target instrument within a mix. This finding suggests the potential utility of frequency domain sparsity as an adaptive strategy to improve auditory perception in individuals with hearing challenges. Furthermore, the higher roll-off points of the target with respect to that of the mix may well serve as an indicator of the energetic masking release of the former from the latter. Many previous studies elaborate upon the reduced frequency selectivity brought on by cochlear hearing loss [36]. These discrepancies usually resulting from broader auditory filters may underpin our findings behind oHI listeners benefiting from sparser mixes with higher spectral contrast and higher roll-off point differences between target and mix.

These higher positive roll-off differences allude to reduced energetic masking of the target by the mix, which may in turn provide an overall benefit in MSA performance irrespective of hearing loss.

## Limitations

In the sample of participants considered in this study, the yNH participants were significantly younger than the oHI participants. Therefore the effect of hearing impairment on MSA performance observed, may also be due to a composite effect of hearing loss and age. Several studies allude to age being a critical determinant in music perception despite hearing loss. In one such study, Bones and Plack [37] showed that aging among normal-hearing listeners brought about a depreciation in neural representation responsible for differentiating between consonant and dissonant chords made up of two notes. Moreover, older participants rated dissonant chords as being more pleasant than consonant chords unlike their younger peers. In another study by Cohrdes et al. [38], there were observable differences in valence and arousal perception brought on by age for musical stimuli but not for non-musical sounds. On an independent note, Goossens et al. [39] showed that speech perception amid a masker has been shown to depreciate for participants above 50 years of age, even though they presented with normal-hearing loss levels up to 4 kHz. However, the same for music perception remains moot. Based on previous research, there is a possibility that our results could see differences if we considered yNH and oHI participants of similar ages. On that note, it is also important to highlight the fact that normal-hearing participants were more musically trained than their hearing-impaired peers. Previous literature compliments the advantages offered by musical training not only in music perception but also perceiving masked speech [40, 41]. Chen et al. [42] showed among children with cochlear implants that early introduction of musical perception training among such children may augment their ability to distinguish pitch differences between two successive Piano tones. Larrouy-Maestri et al. [43] showed that the Gold-MSI musical training scores as that used here, correlated positively with the participants' ability to perceive mistuning or pitch shifts in Lead vocal tracks with respect to that of the accompaniment in pop music. Musicians were also more likely to identify if a complex tone had a mistuned second harmonic [44]. Although musical training offers a plethora of advantages in music and speech perception, within the context of MSA ability, evidence supporting a similar assertion remains rather weak. Case in point, Hake et al. [1] through investigating scene analysis abilities among participants with a wide rage of musical abilities showed that there was but a modest correlation between musical training scores and MSA performance. This finding does not necessarily conflict with our analysis where musical training had no observable effects on the MSA performance. The absence of a significant effect brought on by musical training could therefore be due to the shortcomings brought on by comparatively small and unbalanced sample sizes between yNH and oHI. Nevertheless, controlling for musical training may potentially reduce the performance gap between the participant groups. On the note of classifying hearing impairment, we relied heavily upon the pure-tone audiometry. The area of auditory processing disorder addresses the phenomenon behind participants having difficulties in making out sounds amid a masker despite presenting with normal audiograms [45]. As previous studies in the area suggest the implications of various factors in such shortcomings [46], more customized screening methods aimed at gauging these factors should be used in tandem with audiometry to more accurately assess the physiological and psychological causes of processing disorders impinging on MSA ability.

## Conclusion

In an attempt to understand how music can be remixed for listeners with sensorineural hearing impairment to better facilitate their ability to successfully hear out a target instrument (MSA performance) within a coherent mix of several instruments and vocals, we applied the EQ-transform introduced in our previous study. This transform was used as a means of manipulating the spectral contrast on both the target and the individual tracks making up the mix. Despite having no effect on the objective frequency domain sparsity of the target and the mix measured using the Gini index, the transform did bring about significant and monotonical changes to the mix roll-off points. We assessed MSA performance as a function of hearing loss levels, musical training using the Gold-MSI questionnaire, and the level of spectral manipulation by way of the % EQ-transform. Although the participants reported varying degrees of musical training, its effect on MSA performance was negligible. As shown previously by Hake et al. [1], the performance depended strongly upon the category of the target instrument and to a much lesser extent on the hearing loss levels of the participants. Importantly, our results reveal a notable trend where as levels of hearing loss increased, MSA performance saw a steady decline, reaching even chance-level at hearing thresholds characteristic of moderately severe impairment or worse. This finding is similar to that shown by Hake et al. [1] where listeners with severe to profound hearing loss performed at near chance-levels. Interestingly, the MSA performance among hearing-impaired participants saw a significant improvement for spectrally sparser mixes unlike that among normal-hearing participants which remained robust to changes in mix sparsity. Both participant groups benefited from target roll-offs being larger than that of the mix which may serve as a marker for reduced energetic masking of the former from the latter. Our findings thus far show that, within a musical scene analysis context, spectral manipulations to popular multi-track music may benefit hearing-impaired listeners. Given the complex interplay between hearing loss and various other factors in musical scene analysis, it would be beneficial to formulate auditory models to streamline our understanding of it. As a first step towards creating such models, in a future work, we aim to explore effective models of speech intelligibility and sound quality in their utility to predict performance in the music perception tasks investigated here.

## Author Contributions

**Conceptualization:** Aravindan Joseph Benjamin, Kai Siedenburg.

**Data curation:** Aravindan Joseph Benjamin.

**Formal analysis:** Aravindan Joseph Benjamin.

**Funding acquisition:** Kai Siedenburg.

**Investigation:** Aravindan Joseph Benjamin.

**Methodology:** Aravindan Joseph Benjamin, Kai Siedenburg.

**Project administration:** Kai Siedenburg.

**Software:** Aravindan Joseph Benjamin.

**Supervision:** Kai Siedenburg.

**Validation:** Aravindan Joseph Benjamin, Kai Siedenburg.

**Visualization:** Aravindan Joseph Benjamin, Kai Siedenburg.

**Writing – original draft:** Aravindan Joseph Benjamin.

**Writing – review & editing:** Kai Siedenburg.

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
