## [Decision Letter · Decision Letter 0]

20 Sep 2024

PONE-D-24-19971Effects of spectral manipulations of music mixes on musical scene analysis abilities of hearing-impaired listenersPLOS ONE

Dear Dr. Benjamin,

Thank you for submitting your manuscript to PLOS ONE. After careful consideration, we feel that it has merit but does not fully meet PLOS ONE’s publication criteria as it currently stands. Therefore, we invite you to submit a revised version of the manuscript that addresses the points raised during the review process.

We look forward to receiving your revised manuscript.

Kind regards,

Mohammed Zakariah

Academic Editor

PLOS ONE

Journal Requirements:

2. Thank you for stating the following financial disclosure: Freigeist Fellowship from the Volkswagen Stiftung 

Additional Editor Comments:

Please incorporate the reviewers comments.

Reviewers' comments:

Reviewer's Responses to Questions

**Comments to the Author**

1. Is the manuscript technically sound, and do the data support the conclusions?

Reviewer #1: Partly

Reviewer #2: Partly

2. Has the statistical analysis been performed appropriately and rigorously? 

Reviewer #1: Yes

Reviewer #2: Yes

3. Have the authors made all data underlying the findings in their manuscript fully available?

Reviewer #1: Yes

Reviewer #2: Yes

4. Is the manuscript presented in an intelligible fashion and written in standard English?

Reviewer #1: Yes

Reviewer #2: Yes

5. Review Comments to the Author

Reviewer #1: The problem needs to be adequately explained. What is the motivation of this work?? Is the dataset used in this work a benchmark?? Please show the sample of the data. In the abstract, the results should be discussed in numerical. How accurate was the proposed methodology?? What could be the limitations? Briefly discuss the major significant contributions of this work. explicitly . The methodology should be supported with images. How the flow of the work is goining. What is the architecture of the work. The discussion should be based on recent works done in tis domain.

Reviewer #2: • References are not new. Please cite papers from 2024 and 2023.

• Please cite the latest references in the introduction. This gives the impression that this study focuses on current problems

• The paper had a long introduction, most of which can be transferred to the literature review section. Give a tabular format for this information with some columns like limitations, results, and methodology

• The figures are blur. Please include high-resolution images.

• Please discuss the methodology applied explicitly and what the novelty is.

• What are the performance evaluation metrics/? How are results judged??

• Compare the results with state-of-the-art techniques

• What are the limitations and future scope of the work??

• The references do not look very recent.

6. PLOS authors have the option to publish the peer review history of their article (what does this mean?). If published, this will include your full peer review and any attached files.

Reviewer #1: **Yes: **Dr. Aditi Sharma

Reviewer #2: **Yes: **Rashmi Murgendra Ashtagi

---

## [Author Response · Author response to Decision Letter 0]

11 Nov 2024

In light of Reviewer 1‘s question regarding the database being a benchmark, we wish to mention that the open-source Medeley database used in our study has been used in several peer reviewed studies in the area of music perception, psychology, and processing. Namely, Bannister et al. (2024) used the database in their study titled ‘Muddy, muddled, muffled ? Understanding the perception of audio quality in music by hearing aid users, published in the Frontiers of Psychology to assess audio quality evaluations with listeners with mild to severe hearing loss. Owing to its very recent date of publication, it has not garnered citations so far. The database was also used in another study titled ‘The mistuning perception test : A new measurement instrument, by Larrouy-Maestri et al. (2019), to study perception of mistuning among vocals in popular music. This study published in the Behavior Research Methods has garnered over 40 citations. 

As for Reviewer 2‘s concern regarding the references being old, the area of music perception and particularly scene analysis among listeners with cochlear hearing loss is rather niche and poorly researched. As such, recent research within the purview, especially between 2023-24 is very rare. Most of the studies that provide the psychacoustical basis from which we build our research question were conducted well before year 2020. Nevertheless, we have added relevant studies which are relatively recent (2003 - 2024) in the revised manuscript.

---

## [Decision Letter · Decision Letter 1]

11 Dec 2024

Effects of spectral manipulations of music mixes on musical scene analysis abilities of hearing-impaired listeners

PONE-D-24-19971R1

Dear Dr. Benjamin,

We’re pleased to inform you that your manuscript has been judged scientifically suitable for publication and will be formally accepted for publication once it meets all outstanding technical requirements.

Kind regards,

Mohammed Zakariah

Academic Editor

PLOS ONE

Additional Editor Comments (optional):

Authors have revised the paper based on the reviewers comments and hence the ppaer can accepted for publication.

Reviewers' comments:

Reviewer's Responses to Questions

**Comments to the Author**

1. If the authors have adequately addressed your comments raised in a previous round of review and you feel that this manuscript is now acceptable for publication, you may indicate that here to bypass the “Comments to the Author” section, enter your conflict of interest statement in the “Confidential to Editor” section, and submit your "Accept" recommendation.

Reviewer #2: All comments have been addressed

2. Is the manuscript technically sound, and do the data support the conclusions?

Reviewer #2: Yes

3. Has the statistical analysis been performed appropriately and rigorously? 

Reviewer #2: Yes

4. Have the authors made all data underlying the findings in their manuscript fully available?

Reviewer #2: Yes

5. Is the manuscript presented in an intelligible fashion and written in standard English?

Reviewer #2: Yes

6. Review Comments to the Author

Reviewer #2: (No Response)

7. PLOS authors have the option to publish the peer review history of their article (what does this mean?). If published, this will include your full peer review and any attached files.

Reviewer #2: **Yes: **Rashmi Ashtagi

---

## [Editor Report · Acceptance letter]

30 Dec 2024

PONE-D-24-19971R1 

PLOS ONE

Dear Dr. Benjamin, 

I'm pleased to inform you that your manuscript has been deemed suitable for publication in PLOS ONE. Congratulations! Your manuscript is now being handed over to our production team.

Kind regards, 

on behalf of

Dr. Mohammed Zakariah 

Academic Editor

PLOS ONE